# Bone Cement and Its Anesthetic Complications: A Narrative Review

**DOI:** 10.3390/jcm12062105

**Published:** 2023-03-07

**Authors:** Lou’i Al-Husinat, Basil Jouryyeh, Sarah Al Sharie, Zaid Al Modanat, Ahmad Jurieh, Laith Al Hseinat, Giustino Varrassi

**Affiliations:** 1Department of Clinical Medical Sciences, Faculty of Medicine, Yarmouk University, Irbid 21163, Jordan; zaid.modanat@yu.edu.jo; 2Faculty of Medicine, Yarmouk University, Irbid 21163, Jordan; bassiljoureyah978@gmail.com (B.J.); sarahalsharie2000@gmail.com (S.A.S.); ajurieh@gmail.com (A.J.); 3Department of Orthopaedics, Royal Medical Services, Amman 11855, Jordan; laith_hseinat@hotmail.com; 4Paolo Procacci Foundation, 00193 Rome, Italy; giuvarr@gmail.com

**Keywords:** bone cement implantation syndrome, BCIS, micro-embolism, anesthesia

## Abstract

The concept of bone cement implantation syndrome (BCIS) is not yet fully understood. In patients undergoing cemented hip arthroplasty, it is a significant factor in intraoperative mortality and morbidity. It may also manifest in a milder form postoperatively, resulting in hypoxia and confusion. In the older population, hip replacement surgery is becoming more prevalent. The risks of elderly patients suffering BCIS may be increased due to co-existing conditions. In this article, we present a narrative review of BCIS including its definition, incidence, risk factors, etiology, pathophysiology, clinical features, prevention, and management, all from an anesthetic point of view.

## 1. Introduction

Polymethyl methacrylate (PMMA), also known as bone cement, is a widely used implant fixation material in orthopedic and trauma surgery. In reality, the term “cement” is a misnomer because it refers to a substance that bonds two things together. PMMA, on the other hand, acts as a space-filler, creating a tight space that holds the implant against the bone and thus acts as a “grout”. Bone cement lacks inherent adhesive properties, instead it relies on a mechanical interlock between the irregular bone surface and the prosthesis [1].

PMMA was first developed by the chemist Otto Röhm in 1902 as “Plexiglas”, a glass-like hard material, and it has been used for a variety of applications since then [2]. The invention of modern PMMA bone cements is credited to Degussa and Kulzer (1943), who described the mechanism of polymerization of methyl methacrylate (MMA) at room temperature in the presence of a co-initiator, such as a tertiary aromatic amine. The use of bone cement in orthopedics is widely attributed to the famous English surgeon, Sir John Charnley, who used it for total hip arthroplasty in 1958, and was inspired by his dentist. At first, he used dental acrylic for prosthetic fixation in total hip replacement surgeries (THRs), and in 1965, he started using bone cement that was specifically developed for orthopedic use. This was a significant milestone in the advancement of orthopedic surgical procedures. In the 1970’s, the U.S. Food and Drug Administration (FDA) approved the use of bone cement in hip and knee prosthetic fixation. Since then, bone cement has been used widely in orthopedic procedures [1,3].

Besides its uses in orthopedic surgery, PMMA is used in a different number of applications, including its use in semiconductor research, as a constituent of electric guitar bodies, and as a transparent glass substitute in windows. Furthermore, due to its biocompatibility, it is an important component of intraocular lens implants, dentures, and dental fillings [3].

PMMA is an acrylic polymer composed of two sterile components: a liquid methyl-methacrylate (MMA) monomer and a powered MMA-styrene co-polymer. When the two components are combined, the liquid monomer polymerizes around the prepolymerized powder particles, resulting in hardened PMMA. The process is an exothermic reaction, producing heat [1]. Antibiotics can be incorporated into the polymerized matrix as a soluble powder that is then released into the joint cavity [3]. Interestingly, Qin et al. have published a case report in which BCIS was induced by antibiotic-loaded bone cement covering the infected bone surface [4]. As a result, bone cement serves as a modern drug delivery system, delivering the required drugs directly to the surgical site. In addition, a contrast agent is added to the cement to make it radiopaque, usually zirconium dioxide (ZrO_2_) or barium sulfate (BaSO_4_) [1]. A variety of other additives have been tested in bone cement such as silver-containing nanoparticles which can act as antibacterial agents, bactriocins as an alternative to antibiotics, and vitamin E that showed a positive effect on free radical oxidation and exothermic activity [5,6,7].

## 2. Bone Cement Implantation Syndrome

Bone cement implantation syndrome (BCIS) is a complication associated with the implantation of PMMA and is a significant cause of intraoperative mortality and morbidity. BCIS is commonly associated with patients undergoing cemented hip hemiarthroplasty. However, it also occurs in patients undergoing other arthroplasties such as total hip replacement and knee replacement surgeries. It should be noted that BCIS can also happen, in a milder form, in the postoperative period [8,9].

Although the syndrome is poorly understood and has no standard definition, its clinical presentation is mostly characterized by hypoxia [10,11], systemic hypotension [10,12], pulmonary hypertension [12,13], cardiac arrhythmias [14], loss of consciousness and eventually cardiac arrest [15,16]. In different published articles, these clinical features were reported to occur at one of the following time points during surgery, which might be cementation, prosthesis insertion, joint reduction or, rarely, limb tourniquet deflation in a patient undergoing cemented arthroplasty [8,17,18,19].

In 2009, Donaldson et al. [8] proposed the first classification system for BCIS according to its severity, which was as follows:Grade 1: Moderate hypoxia [arterial oxygen saturation (SpO_2_) < 94%] or hypotension [fall in systolic blood pressure (SBP) > 20%].Grade 2: Severe hypoxia (SpO_2_ < 88%) or hypotension (fall in SBP > 40%) or unexpected loss of consciousness.Grade 3: Cardiovascular collapse requiring cardiopulmonary resuscitation (CPR).

### 2.1. Incidence

The true incidence of BCIS is unknown due to the wide range and ambiguity of its symptoms [8]. However, recently, some studies have investigated the incidence of BCIS in patients undergoing different arthroplasties. In 2020, Rassir et al. [20] carried out a retrospective observational study in which they investigated the incidence of BCIS in 3294 patients undergoing cemented arthroplasties for any reason, and the overall incidence of BCIS among all included arthroplasties was 26% (845 of 3294). In hip hemiarthroplasty, the incidence was 31% (282 of 915), and in total hip arthroplasty (THA), it was 24% (165 of 677). Additionally, the incidence was 28% (210 of 765) in total knee arthroplasty (TKA), 20% (113 of 558) in unicondylar knee arthroplasty, 23% (47 of 206) in revision arthroplasty, and 16% (28 of 173) in shoulder arthroplasty (16). In another retrospective study of 1016 patients undergoing cemented hemiarthroplasty for femoral fractures, the authors reported an overall rate of any grade of BCIS of 28%; the rates of BCIS grade 1, 2, and 3 were reported at 21.0%, 5.1 %, and 1.7%, respectively [9]. Furthermore, in a more recent similar retrospective observational study, Weingärtner et al. [17] investigated the incidence of BCIS in 208 patients undergoing cemented hemiarthroplasty for a proximal femur fracture. In total, 37% of the patients had BCIS symptoms.

Apart from the aforementioned similar results reporting BCIS incidence in cemented hemiarthroplasty (31% by Rassir et al., 37% by Weingärtner et al., and 28% by Olsen et al.), Schwarzkopf et al. [21] reported an incidence of BCIS of 74% in cancer patients undergoing cemented hip arthroplasty, which is almost 2.5 times as much as the BCIS incidence in non-cancer patients. This is mostly explained by the hypercoagulability status of cancer patients and their increased tendency to develop emboli [22].

### 2.2. Clinical Manifestations

The clinical manifestations of BCIS range from mild to severe. Patients may experience transient hypotension and hypoxia, as well as cardiac arrhythmias and, in the most severe scenario, cardiac arrest. Most studies explain these changes by the presence of right ventricular failure secondary to an increase in pulmonary vascular resistance (PVR). This increase in the afterload dilates the thin-walled right ventricle and shifts the interventricular septum to the left, which, in turn, causes a decrease in left ventricular compliance, reduced left ventricular filling, and cardiac output (CO). Reasons underlying the increase in PVR are still not clear. It could be caused by the systemic absorption of the volatile monomer or by the deposition of cement or fat emboli [3]. Clinical signs of BCIS, as well as a sudden decrease in end-tidal CO_2_ (ETCO_2_), are strong indicators of BCIS; however, a computed tomographic (CT) scan is required to make a definitive diagnosis [23].

### 2.3. Etiology/Pathophysiology

The etiology and pathophysiology of BCIS are not yet understood. However, earlier theories focused on circulating MMA monomers, but new evidence suggests an embolus-mediated model [24]. In addition, there are other several mechanisms that have been proposed such as complement activation, histamine release, and a multimodal model [8].

#### 2.3.1. Monomer-Mediated Model

Early studies theorized that circulating methyl methacrylate (MMA) monomers cause vasodilatation and, thus, are responsible for the cardiovascular and pulmonary effects seen in BCIS. However, this was only demonstrated in vitro. In vivo, a number of animal studies have shown that the plasma MMA concentration after cemented hip arthroplasty is significantly lower than the concentration required to cause pulmonary or cardiovascular effects [8]. As a consequence, subsequent studies have focused more on the embolic model in their attempt to explain the pathophysiology underlying BCIS [3,8,24].

#### 2.3.2. Embolus-Mediated Model

Several postmortem and echocardiographic studies in the literature suggested this model. Postmortem studies on animals and humans have revealed pulmonary embolization and echocardiographic studies detected the presence of emboli in the right atrium, right ventricle and pulmonary artery intraoperatively. It has been established that these emboli contain fat, marrow, air, bone particles, cement particles, and aggregates of platelets and fibrin. The effects of these emboli are believed to be either due to mechanical effects, mediator effects or both [8,24]. The pathophysiology of this model is shown in Figure 1.

##### Mechanical Effects

It is theorized that during cementation and prosthesis insertion, high intramedullary pressures (>300 mm hg) develop and cause embolization, and this has been proven in both in vitro and in vivo studies. The exothermic reaction of the cement expands the space between the prosthesis and the bone, trapping air and debris and forcing it into the circulation. Emboli then travel to the right atrium, right ventricle, and to the pulmonary vasculature causing the manifestations of BCIS [8,24]. In addition, in a study by Koessler et al. [25], 120 patients underwent THR procedures using conventional vs. modified cementing techniques. The modification involved inserting a suction catheter into the proximal femur to reduce the increase in intramedullary pressure during prosthesis insertion. Embolic events were imaged during prosthetic stem insertion in 93.4% of patients using the conventional cementing technique and 13.4% of patients using the modified technique. However, although embolic spreading may explain the manifestations of BCIS, research has not yet established a link between the embolic load and the cardiovascular compromise accompanying BCIS [26].

##### Mediator Effects

In addition to mechanically obstructing the pulmonary vasculature, emboli act by other means to increase PVR. First, emboli may cause endothelial damage which leads to the release of endothelin factor-1 from the pulmonary vessels’ endothelium, which causes vasoconstriction, or they may mechanically stimulate the pulmonary vasculature and cause reflex vasoconstriction. Second, it has been theorized that the embolic material itself may release vasoactive or pro-inflammatory mediators such as thrombin and tissue thromboplastin that directly increase PVR, or act indirectly by promoting the release of additional mediators that increase PVR. Emboli obstructing the pulmonary vessels and the pulmonary vasoconstriction induced by the mediators are responsible for the ventilation perfusion (V/Q) mismatch and the development of hypoxemia. There are other mediators that cause a reduction in the systemic vascular resistance (SVR) such as 6-keto PGF 1α and tissue thromboplastin, and that is through the release of secondary mediators such as adenine nucleotides. Increased PVR in the presence of decreased RV preload, due to a reduction in the SVR, results in a significant decrease in cardiac output (CO). As the CO declines, hypotension worsens [8,24]. Medullary lavage prior to cement insertion significantly reduces the release of some of these mediators [27].

Among all proposed hypotheses explaining BCIS, the embolic model is the most dominant one in the literature. However, it has some flaws. First, this model does not explain all observed phenomena. Second, embolization is not always associated with hemodynamic changes, and the degree of embolism has a poor correlation with the degree of hypotension or hypoxemia [28]. Transesophageal echocardiography (TEE) results show that while embolic events are frequently observed, most patients tolerate them well. This has led current researchers to believe that micro embolism, rather than being the primary mechanism of BCIS, may be a contributing factor [3,8,28].

#### 2.3.3. Histamine Release and Hypersensitivity

BCIS and anaphylaxis share a few clinical features. Patients with hypotension undergoing cemented hip arthroplasty have been found to have higher plasma histamine concentrations [29]. There are no studies that confirm the cause of histamine release, whether it is a direct action of the cement or an IgE-mediated process. One study found that blocking histamine receptors with clemastine and cimetidine (H1 and H2 antagonists) provided modest protection [29]; however, those findings were not replicated in two subsequent studies [30,31].

#### 2.3.4. Complement Activation

The complement-derived peptides C3a and C5a have long been known to have anaphylatoxic properties. They are potent vasoconstrictors and bronchoconstrictors [32]. Under normal conditions, these peptides complement our immune system; however, when it comes to BCIS, increased levels of C3a and C5a result in smooth muscle contraction, increased vascular permeability, and histamine release via either a direct effect of MMA or a hypersensitivity reaction. Clinically, it presents as pulmonary vasoconstriction, systemic hypotension, and desaturation [3,8]. In a later study, the authors failed to identify complement activation [33].

#### 2.3.5. Multimodal Model

The most likely explanation for BCIS is a combination of the theorized models in conjunction with an individual physiologic response. Preexisting comorbidities, surgical technique, and the proposed surgery may all affect a patient’s reaction to bone cement [8,24].

### 2.4. Relationship with Anesthesia

Because most BCIS documented cases occurred in patients undergoing hip hemiarthroplasty [9], there has been an ongoing debate in the literature regarding the best anesthetic technique in patients undergoing this type of surgery. Both regional and general anesthesia are possible options for hip fracture surgery, but which technique produces the best results is still debated [34,35].

We found several studies in the literature suggesting that there may be some advantages to using spinal anesthesia over general anesthesia for this kind of surgery. It is proposed that regional anesthesia may improve outcomes by avoiding intubation and mechanical ventilation, reducing blood loss, and improving postoperative analgesia. In contrast, general anesthesia may provide better hemodynamic stability than regional anesthesia [36]. In a retrospective nationwide population-based study, Ahn et al. [37] concluded that in 96,289 elderly patients who underwent hip fracture surgery, regional anesthesia was associated with better outcomes than general anesthesia in terms of mortality, delirium, intensive care unit admission and ventilatory care. In another retrospective study, Chu et al. [38] found that neuraxial anesthesia was associated with a lower risk of adverse outcomes compared to general anesthesia. White and Griffiths [39], with a sample of 5029 patients, 760 (15.1%) of whom experienced BCIS, observed a significant statistical reduction in BCIS prevalence when comparing patients who received only spinal anesthesia to those who received only general anesthesia. However, in the 2371 patients who were administered general anesthesia, without additional spinal anesthesia, there was no statistical difference in BCIS prevalence between patients who were mechanically ventilated and those who were self-ventilating. Neumann et al. [34] carried out a retrospective study, gathering data from 18,158 patients at 126 hospitals, and they found no differences in unadjusted in-hospital mortality between the groups studied. However, the adjusted analysis confirmed that the group undergoing regional anesthesia had a 29% reduction in perioperative mortality and lower odds of pulmonary complications when compared to the group undergoing general anesthesia. The necessity for multiple variable adjustments suggests that general anesthesia and spinal anesthesia definitions may be too broad for practical clinical recommendations. Future research on best anesthetic practice should look into the best method for general anesthesia or the best method for spinal anesthesia [39]. The mentioned findings regarding mortality and complications [34] are similar in direction and magnitude to those of Radcliff et al. [40], who in a sample of 5683 patients undergoing hip fracture surgery, found a higher risk of mortality and complications in patients undergoing general anesthesia when compared to patients undergoing regional anesthesia.

To our knowledge, there is only one Cochrane database review in the literature supporting the use of spinal anesthesia over general anesthesia in hip fracture surgery in adults [36]. However, in an updated Cochrane database review, the authors found no differences in general and spinal anesthesia when they compared data from 3231 participants in the areas of mortality, pneumonia, myocardial infarction, cerebrovascular accident, acute confusional state, or a patient’s return home [35].

Although we found several studies that supported the use of spinal anesthesia, most of them found no difference regarding the best type of anesthetic technique in hip arthroplasty. In a recent study published in 2022, the authors concluded that spinal and general anesthesia both achieve similar outcomes for patients undergoing hip fracture surgery and they observed no differences in mortality between the two anesthetic groups [41]. In another recent study, Ana et al. [42] identified 562 patients who had hip fracture surgery, 361 of whom were given general anesthesia and 201 of whom were given regional anesthesia. There was no statistically significant difference in the risk of perioperative and 30-day mortality in the adjusted analysis. Moreover, an 11-year retrospective study of 7164 patients and a 5-year retrospective study of 73,284 patients both found no superior anesthetic technique in terms of perioperative mortality [43,44].

Furthermore, a systematic review and meta-analysis of randomized and nonrandomized studies published between January 2000 and July 2017 found no significant difference in 30-day mortality, despite a small statistically significant difference in length of stay, which is unlikely to be clinically significant, favoring regional anesthesia [45]. More recently, another systematic review and meta-analysis of randomized controlled trials published between 2000 and February 2022 showed no significant difference between spinal anesthesia and general anesthesia, except for a reduced risk of acute kidney injury (AKI) in patients who underwent spinal anesthesia [46].

In conclusion, the best anesthetic technique for hip arthroplasty is not yet identified and it follows that the best practices for preventing the potential, but rare, adverse effects of BCIS have not yet been defined. However, a thorough preoperative assessment of the patient and identification of individual patient risk factors should guide the anesthetist in developing an anesthetic plan that optimizes each individual’s cardiopulmonary health [24].

### 2.5. Prevention of BCIS

Preoperative evaluation and continuous intraoperative vigilance are critical in the prevention and management of BCIS due to a lack of consensus in the literature regarding the best anesthetic technique for cemented bone surgery (21). BCIS prevention begins with identifying high-risk patients during the preoperative assessment and communication between surgical, anesthetic, and medical providers regarding the selection of the type of prosthesis, surgical procedure, and techniques to minimize the risk of cardiovascular complications for high-risk patients with multiple or severe risk factors or comorbidities. Assessing and optimizing patients’ cardiovascular reserve prior to surgery is also critical in the prevention of BCIS [3,24,47].

Patient risk factors correlated with the development of BCIS after cemented total hip replacement surgeries (THRs) include an age greater than 65 years old, male sex, ASA score of three or over, severe cardiopulmonary disease, pre-existing pulmonary hypertension, the use of diuretics or warfarin, poor pre-existing physical reserve, malignant tumors, porous bone quality (osteoporosis), bony metastasis, mainly pathological hip fractures and intertrochanteric fractures [3,8,9]. Chronic obstructive pulmonary disease (COPD), which is frequently complicated by pulmonary hypertension and increased PVR, is an additional risk factor. In the case of BCIS, underlying COPD contributes to hypoxia, acidemia, and the overall inflammatory response [9].

Additionally, there are surgical risk factors that are linked to BCIS. Patients who have previously un-instrumented femoral canals are more likely to develop the syndrome than those who have revision surgery. Moreover, the use of a long-stem femoral component raises the chances of developing BCIS [48,49].

As previously mentioned, there is no conclusive evidence that the anesthetic technique affects the severity of BCIS. However, inhalational agents have been observed to have more profound hemodynamic effects in the presence of BCIS. In addition, nitrous oxide should be avoided in patients with BCIS to avoid air embolism exacerbation [3,8,23]. During arthroplasty, regional anesthesia with little or no sedation allows for the evaluation of mental status and dyspnea [50,51]. Nevertheless, in the event of BCIS, this should be balanced against the need for a secure airway and maximum ventilation [24].

Anemia is common in adults over the age of 65 (17%), and the incidence is even higher in the hip fracture population. Thus, preoperative and intraoperative blood and fluid restoration of the patient should be considered, and intravascular fluid volume should be kept as close to normal as possible [52].

#### 2.5.1. Monitoring

A sudden and unexplained decrease in end-tidal carbon dioxide (ETCO_2_) is an early sign of clinically significant BCIS, and whether the patient is under general or regional anesthesia, capnography is indicated. Hypoxia is also a key symptom of BCIS, making pulse oximetry indispensable [51]. Consequently, inspired oxygen concentration should be increased in all patients at the time of reaming, cementation, reduction of the joint and deflation of tourniquet [3].

Bradycardia and hypotension are early hemodynamic indicators of BCIS. As a result, it is ideal to use continuous electrocardiography (ECG) and keep systolic blood pressure within 20% of baseline [50]. Because of the hemodynamic changes seen in BCIS, it is best to consider high levels of hemodynamic monitoring in high-risk patients in the OR [8]. First, invasive arterial blood pressure (ABP) monitoring may be recommended in the presence of a possible hemodynamic instability, an expected need for frequent arterial blood gas analysis, an expected use of vasopressors, or in a patient with multiple BCIS risk factors [51]. Second, central venous pressure monitoring (CVP) may help with volume optimization; however, if the patient has BCIS and elevated pulmonary artery pressures, central venous pressure monitoring might become ineffective [24]. Lastly, intraoperative cardiac output (CO) monitoring has been recommended in patients who have one or more risk factors for BCIS. CO monitoring can be in the form of a semi-invasive transesophageal Doppler monitor or invasive CO monitor (pulmonary artery flotation catheter). In a study involving 20 patients undergoing hip arthroplasty, Clark et al. [12] found that transesophageal Doppler monitoring outperformed standard hemodynamic monitoring in detecting cardiovascular changes during cementation, recommending its use in high-risk patients. The use of intraoperative transesophageal Doppler has been shown to assist in the early detection of cardiovascular changes around the time of cementation, improve fluid management, and decrease the incidence of postoperative cardiopulmonary complications in hip surgery [3]. Pulmonary artery flotation catheter and transesophageal echocardiography should also be considered in patients who are at high risk of developing BCIS. Nevertheless, their use is limited by their availability and the expertise required to use them [3].

#### 2.5.2. Surgery’s Role

A variety of surgical measures may be used to reduce the risk of BCIS. A cement-free prosthesis appears to be the most likely option for avoiding BCIS. However, when researchers compared cemented arthroplasty to non-cemented arthroplasty for hip fractures, cemented arthroplasty resulted in less residual pain, better mobility, and a lower need for revisions [53]. If cement is considered necessary, using a short-stem prosthesis, low-viscosity cement, lavage of the intramedullary canal prior to prosthesis insertion, and achieving hemostasis before implanting the prosthetic joint can reduce the patient’s risk of BCIS [3,23].

Cemented hip arthroplasty has been shown to have significantly higher perioperative mortality than uncemented hip arthroplasty. However, after one year, mortality rates reversed, with patients who had cemented hip arthroplasty having a better chance of survival [54,55].

The method of preparing bone cement is also relevant to BCIS. When mixing the cement, using a bone-vacuum cementing technique, rather than mixing the cement at atmospheric pressure, reduces micropores and macropores. Decreased cement porosity reduces the embolic load of both mechanical and mediator-driven particles. Vacuum mixing also increases cement strength, ensures consistent cement results, reduces the risk of cement loosening or cracking, and limits exposure to cement fumes [56,57]. Finally, retrograde insertion of bone cement (distal to proximal) using a cement gun helps to compartmentalize bone marrow contents, providing a uniform lower intramedullary pressure and reducing the risk of BCIS [3,23].

### 2.6. Management

BCIS is a reversible, time-limited phenomenon, according to several studies. Pulmonary artery pressure can return to normal within 24 h, and non-diseased hearts can recover in minutes to hours [3]. This means that the early detection of BCIS combined with aggressive supportive therapy is key to preventing the complications of this syndrome [24]. If BCIS is suspected, immediate resuscitation should be based on general principles. Initially, the anesthetist should secure the airway, increase the inspired oxygen concentration to 100%, and accurately monitor the ETCO_2_ [3,50,51]. In grade 3 BCIS, cardiopulmonary resuscitation should be initiated. If the anesthetist is using propofol infusion, it is advised to discontinue or decrease its use until the patients’ hemodynamic level is stable [58].

It has been suggested that in the context of BCIS, cardiovascular collapse should be treated as RV failure [59,60]. This includes the administration of intravenous fluid therapy to maintain normovolemia and avoid hypotension, the use of pulmonary vasodilators (inhaled nitric oxide and prostaglandin E1) to decrease the pulmonary artery pressure if hypoxemia and right ventricular dysfunction worsens, and the use of inotropes (dobutamine and milrinone) to preserve right ventricular contractility [3,24]. Additionally, to maintain hemodynamic stability, direct-acting α-1 agonists (epinephrine, norepinephrine) may be administered [8,50,51,61]. Additionally, in cases of bradycardia and hypotension, ephedrine, an α- and β-adrenergic agonist, may be given; however, giving any sympathomimetics without addressing pulmonary vasoconstriction may result in rapidly increased right ventricular preload and acute right ventricular failure [58]. Lastly, glycopyrrolate, an anticholinergic, can be used as a preventative or rescue medication to counteract an intracardiac chemoreceptor-mediated vasovagal reaction [58].

It is theorized that one of the reasons behind pulmonary vasoconstriction in BCIS is serotonin. That being said, ondansetron, a 5-hydroxytryptamine (5-HT3) receptor antagonist, was given as prevention, both preoperatively and intraoperatively [58]. Similarly, histamine receptor blockade by H1 and/or H2 antagonists (clemastine and cimetidine) has been used, both preoperatively and intraoperatively, to block the direct effects of cement on histamine release or to block the IgE-mediated process of hypersensitivity (anaphylaxis) [8].

Corticosteroids have been used to treat any inflammatory or anaphylactoid reactions associated with BCIS, but they should also be considered prior to surgery if the patient has any preexisting hyper-allergic, mast-cell activation, or inflammatory comorbidities [51,62]. The administration of pre- or intraoperative NSAIDs would also be helpful to prevent inflammatory responses [63,64]. Furthermore, to correct metabolic acidosis, sodium bicarbonate may be administered [51].

Intraoperative fluid resuscitation with blood or blood products should be taken into account because pre-existing anemia may exacerbate the circulation’s reduced ability to deliver oxygen to vital organs in BCIS [52]. Moreover, central venous pressure lines may help and guide in the administration of fluid therapy, vasopressors, or inotropes. Following surgery, the patient should be managed in an intensive care unit setting [3].

In a study by Bökeler et al. investigating the effectiveness of a modified third generation cementation technique and the vacuum mixing of bone cement on BCIS, the authors concluded that such a technique was associated with a statistically significant reduction in BCIS incidence and, thus, mortality rates [65].

## 3. Conclusions

We have reviewed the prevalence and clinical characteristics of BCIS and showed one of the suggested formal definitions and scales of severity for BCIS. We have investigated several hypotheses about the pathophysiology of BCIS, including the embolic model, the monomer-mediated model, and the complement-mediated and histamine models. Rather than fixating on one specific mechanism, we have explored all the mechanisms suggested in the literature and concluded that the probable etiology behind this syndrome may be thought of as multifactorial. Even though most of the cement techniques are secure, we have identified distinct patient populations who are particularly vulnerable to developing BCIS and have offered some recommendations to reduce this risk. We advise against using a cemented prosthesis in patients who have a high likelihood of developing BCIS, unless there are important orthopedic factors. The clinical management, prevention, and monitoring of a patients developing BCIS has also been covered, with a focus on the use of direct-acting alpha agonist medications.

## Figures and Tables

**Figure 1 jcm-12-02105-f001:**
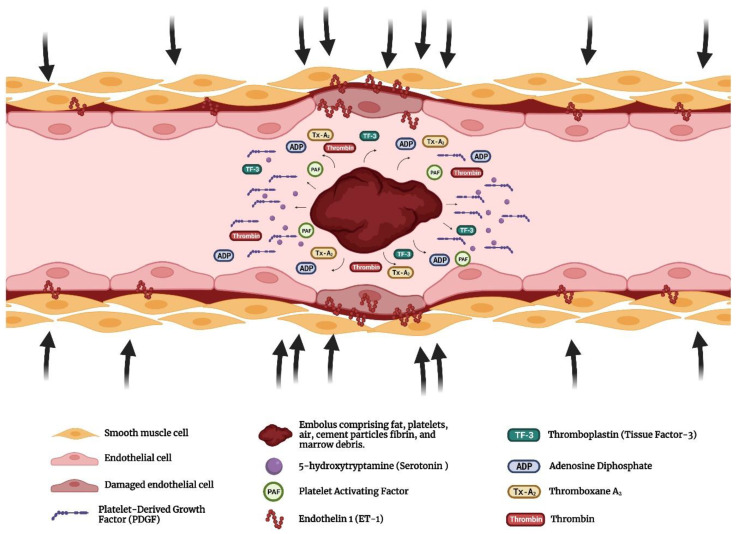
Pulmonary vessel with embolus comprising fat, platelets, fibrin, and marrow debris. The effects of such embolus can be either due to mechanical effects, mediator effects or both.

## Data Availability

All data are available in the presented manuscript.

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
