# Peer review of "Bone Cement and Its Anesthetic Complications: A Narrative Review"

_jcm, 2023, doi:10.3390/jcm12062105_

Round 1

Reviewer 1 Report

This is a well written narrative review on this topic. The writing is clear and appears to be very comprehensive. As such a review I feel it is appropriate for publication as it provide a useful guidance

The article could have a paragraph suggestion future areas of research. ? steroids or other drugs before cementing. ? finding a less toxic cement without the polymethylacrylate component. Comparing a short stem prosthesis versus a standard stem. However this is not essential. 

Author Response

Dear esteemed reviewer, 

We thank you for your comments.

As requested, we have added paragraphs discussing future areas of research, the use of steroids or other drugs before cementing, and the use of alternative, less toxic cement without the polymethylacrylate component.

Reviewer 2 Report

Check the use of subscripts throughout the document (line 114 CO2, Line 52 ZrO2 and BaSO4  ).

Since from line 52 of the paper the authors mention that different antibiotics and additives have been incorporated in the cements formulations, it is important to mention in the paper if any incidence of these variations in the formulations on BCIS has been reported.

More recent information on studies on the subject should be included. The search in Scopus with the equation "Bone Cement Implantation Syndrome" resulted in 127 documents. Of these, 36% (46 papers) corresponded to document results from the last 5 years.

Authors should incorporate tables and/or figures in the paper to help summarize the information presented.

Author Response

Dear esteemed reviewer,

We thank you for your comments.

As requested, subscripts were checked and edited as suitable in the manuscript.

As requested, we have discussed  a study in which antibiotics and additives have been incorporated in cements formulations.

As requested, we have added and discussed more recent studies of BCIS.

The manuscript has a figure illustrating the full mechanism by which BCIS occurs. Regarding tables, most of the relevant published literature of the topic was of non-homogenous variables, and that was why we were not able to display data in tables but imbedded within the text of the manuscript instead.